**Data Availability Statement:** The data from the study cannot be provided to the public as it

# Implementing a physical activity project for people with dementia in Germany–Identification of barriers and facilitator using consolidated framework for implementation research (CFIR): A qualitative study

**Maria Isabel Cardona**[1]*, **Jessica Monsees**[1], **Tim Schmachtenberg**[2], **Anna Grünewald**[1], **Jochen René Thyrian**[1,3]

**1** German Center for Neurodegenerative Diseases (DZNE), Site Rostock/Greifswald, Greifswald, Germany, **2** Department of General Practice, University Medical Center Goettingen, Goettingen, Germany, **3** Institute for Community Medicine, University Medicine Greifswald, Greifswald, Germany

* mariaisabel.cardona@dzne.de

## Abstract

### Background

Despite physical activity (PA) health benefits, people with dementia (PwD) continue to report low levels of PA engagement compared with healthy older adults. Evidencing that PA initiatives still not reflect effective practice and outcomes. Previous studies have shown that several factors can mediate PA initiatives implementation in this population. However, most prior research have not use implementation science frameworks to outline in-depth barriers and facilitators that enables improved PA strategies in PwD. Therefore, a more holistic understanding of mediating factors is still needed.

### Objective

To identify multilevel barriers and facilitator factors, applying the Consolidated Framework for Implementation Research (CFIR) to orient a systematic evaluation of one PA project in PwD and provide evidence-based evaluation results to enhance PA implementation efforts for PwD.

### Method

A qualitative study implemented in 4 German sports associations that applied a PA project for PwD. A total of 13 semi-structured interviews were conducted with 21 participants, project leaders (PLs) and sports trainers (STs). The Consolidated Framework for Implementation Research (CFIR) was used as an evaluation framework to orient both the data collection and analysis.

includes confidential information obtained from the qualitative interviews. Also, because of ethical regulations, complete interview transcripts cannot be provided. However, some transcript extracts are included in the manuscript. On request, a de-identified list of codes or meaning units can be provided, subject to approval by the DOSB steering committee. For data access, contact the Project management "Sport moves people with dementia" via email: nuessler@dosb.de.

**Funding:** The author(s) received no specific funding for this work.

**Competing interests:** The authors have declared that no competing interests exist.

**Abbreviations:** PA, Physical Activity; PwD, Persons with Dementia; CFIR, Consolidated Framework for Implementation Research; DAlzG, German Alzheimer's Association; DOSB, Deutscher Olympischer Sportbund; PLs, Project Leaders; STs, Sport Trainers; QCA, Conventional Qualitative Content Analysis.

## Results

A total of 13 interviews were conducted with 21 participants. The CFIR guided the identification of barriers and facilitating factors that need to be targeted at different levels for successful implementation. Barriers were identified, especially in the external level, as more solid networks and funding for sustainable proposals are still needed. Other barriers were low participation rates, stigma around the disease and the COVID 19 pandemic. On an individual and structural level facilitators were found like motivated appointed leaders, established planning process, and external organizations supporting sports associations in the implementation.

## Conclusion

Sports projects for PwD can benefit from structuring their interventions based on the CFIR framework as it helps identify multilevel factors that may influence their success and promote PA among PwD. Future efforts should continue working on implementing frameworks that facilitate and reduce the complexity of implementing sustainable PA projects for PwD.

## Introduction

Worldwide dementia cases will triple by 2050, from 57.4 million cases in 2019 to 152.8 million cases in 2050 [1]. Dementia has considerable implications for individuals' cognition, function and behavior [2]. Mainly since it is a disease characterized by a progressive loss of memory and cognitive functions [3] accompanied by restrictions in physical functioning involving reduced muscle strength, balance, and mobility [4]. Thus, the public health sector has endeavored to detect potential strategies that might delay the onset and progression of the disease [3]. There is increasing agreement that physical activity (PA) is a beneficial non-pharmacological treatment in prevention and dementia care management that can lead to enhanced cognitive function, independent functioning and psychological health [5–7]. In order to obtain such positive health benefits, according to the World Health Organization (WHO), people over 65 should regularly engage in moderate-intensity aerobic PA for 150 minutes per week [8].

Despite the acknowledged benefits of PA, a gap still prevails between WHO recommendations and actual implementation, as in many cases, PA interventions still do not reflect effective practice and outcomes. Three-quarters of women and three-fifths of men aged 65 years and older engage in less than 150 min of aerobic PA per week, and half of them engage in less than one day per week of aerobic PA [9]. In particular, sedentary time is significantly higher in persons with dementia (PwD) compared to healthy older adults [10]. Given these low levels of PA, several research findings indicate facilitators and barriers to PA participation among PwD For instance, the barriers included bio-medical reasons, physical ability, reduced mobility and pain, cognitive impairments, mental wellbeing, relationship dynamics, socioeconomic reasons, difficulties with guidance, reliance on their care partner, low intrinsic motivation, poor understanding and stigma against dementia, changes in the social and physical environment, poor access to exercise providers and exercise opportunities that met needs and preferences [11–14]. On the other hand, noticeable facilitators for PA were motivation to maintain physical and mental health, participation in preferred PA options, support and guidance for PA, access to convenient, enjoyable, inclusive, and personalized PA options, care partners, social engagement, perceived PA benefits, emotional and physical well-being, social connectedness,

confidence improvements, family support, therapist support, and tailored, fun and flexible group dynamics [11–14]. Other factors mentioned in the literature mediating PA engagement among this population include environmental and community support [12, 15]. Thus, organizations, stakeholders, local planners, communities and health care professionals have the potential to encourage PA while building physical environments, guaranteeing accessible PA venues, and designing and sustaining sport strategies [16]. In this way, there is growing evidence that highlights PA mediator factors that may influence implementation and thus PA participation within the social-ecological model in PwD [13, 17]. Particularly since numerous interpersonal, organizational, community and public policy factors mediate PA engagement and adherence, making it a dynamic and multifaceted process [13, 17, 18].

In spite of the existing evidence [11–14] on the potential barriers and facilitators mediating PA outcomes in PwD, prior research did not provide a scientific angle using frameworks for implementation research. According to the most recent systematic reviews published in 2016 [13]and 2018 [14], studies synthesized barriers and facilitators to PA adherence in PwD, mainly including qualitative methodologies using data collection strategies like interviews and focus groups. However, to narrow the gap from research findings to practice, a more comprehensive insight into the barriers and facilitators is required to bring more effective implementation strategies [19]. For this purpose, a number of theories and models have emerged to indicate factors influencing effective implementation, considering challenging transformations within the healthcare practice, where multiple levels of potential barriers need to be considered [19]. For example, particular features of the practitioners and individuals concerned and the context in terms of the societal, institutional, economic and political environment [20].

One widely applied science framework is the Consolidated Framework for Implementation Research (CFIR) [21–23], which is used as a guiding tool for the multilevel assessment of implementation settings to detect facilitators and barriers to the successful implementation of interventions [22, 23]. Furthermore, it offers a set of standardized and comprehensive implementation constructs, which account for characteristics that may influence implementation and thus play an essential role in whether or not interventions are successful [22, 23]. Thus, this model groups these constructs into five domains: (1) Intervention characteristics (characteristics of the initiative likely to affect implementation); (2) Inner setting (organizational characteristics likely to affect implementation); (3) Outer setting (environmental characteristics likely to affect implementation); (4) Characteristics of individuals (persons engaged in the initiative likely to affect implementation); and (5) Implementation process (planning, executing, evaluating the intervention) [22, 23].

Regardless of the existence of such frameworks, their value to the field of dementia care and PA remains limited [24]. Therefore, there is a need for further research that utilizes implementation science frameworks to outline in-depth barriers and facilitators of the intended change in practice at different levels to clarify the potential drivers of change and the challenging implementation dynamics and enable improved dementia care PA initiatives outcomes [19, 24].

Given the limited use of an application science framework in this particular field of research, this study aims to use the CFIR to identify and synthesize facilitators and barriers when implementing the project "Sports moves people with dementia" (Sport bewegt Menschen mit Demenz) [25], a project initiated by the German Olympic Sports Confederation (DOSB) in cooperation with the German Alzheimer's Society and funded by the Federal Ministry for Family Affairs, Senior Citizens, Women and Youth to introduce new sports offers for people with dementia and their relatives in German sports clubs. In this way, the CFIR framework allowed for synthesizing barriers and facilitators for PA initiatives implementations in PwD, which could provide comprehensive, supportive and scientific guidance for healthcare

policymakers and practitioners to improve future implementation efforts of PA strategies in different settings.

## Methods

Qualitative methods are implemented to recognize crucial multilevel barriers and facilitators in the implementation of a PA project among PwD based on the experience of the subjects involved in the study, and to elicit perceptions towards provision of PA initiatives for PwD within German sports associations.

### Ethic statement

As the study was part of the internal DOSB project evaluation, did not include participants that were patients (PwD), anonymized participants' responses, and did not require physical contact that implied any risk of discomfort or inconvenience to participants, ethical approval from a committee was not sought. Instead, the study was presented in detail to the steering committee of the DOSB project "sports moves people with dementia" (Sport bewegt Menschen mit Demenz). This way, the DOSB steering committee members decided whether to allow this study. The DOSB is a recognized entity with extensive experience in complying with the guidelines and ethical standards of its sports projects, ensuring the consent, confidentiality and anonymity of those participating. After study approval, members of the DOSB steering committee informed project staff about our study, and those interested in participating could do it voluntarily. Interested participants received written information by e-mail about taking part in an interview study. This invitation specified that participation would be anonymized, with no possibility of identification from third parties, with the possibility of ending the interview at any moment with no need for explanation and with the assurance that this would not be reported to those responsible for the project. Participants' consent was informed orally. Written consent was not obtained to avoid further documentation that could compromise the participants' privacy. Verbal informed consent was given at the beginning of each interview.

### Setting

The Federal Government of Germany developed the National Dementia Strategy [26] under the leadership of the Federal Ministry of Health, the Federal Ministry for Family Affairs, Senior Citizens, Women and Youth and the German Alzheimer's Association (DAlzG). One of the defined areas of action in this strategy includes social participation, which seeks to encourage local authorities to provide more dementia-sensitive public places, including sporting opportunities [26]. In particular, one project that is in accordance with the National Dementia Strategy and reflects this increased level of commitment promoting PA among PwD is the large-scale project *"Sport bewegt Menschen mit Demenz" (Sports moves people with dementia)* [25]. This project aimed to test different sports initiatives among PwD from October 2020 to June 2022 within four German sports associations: The German Sports Associations of Lower Saxony, the German Sports Associations of North Rhine-Westphalia, the German Table Tennis Association, and the German Gymnastics Association. Each association offered different approaches to promote PA among the target group. For instance, the sports associations of Lower Saxony implemented educational programs for sports trainers, expanded exercise offers for PwD and their relatives, and promoted sports networking on a national and regional level [25, 27]. The sports associations of North Rhine-Westphalia based on current offers developed sports initiatives for individuals in the early stage of dementia where they could participate during 12-week in sports activities including accompanied bike rides, canoe trips, walking and hiking, gymnastics, dance and music, and games [25, 28]. The German Table Tennis

Association offered PA opportunities where PwD practiced coordination and resistance exercises while playing table tennis, trained cognitive skills while learning different movements, and enhanced social skills while playing with others. Some modifications were made to the elements used in table tennis to facilitate participation; for example, bigger balls were used. It was also possible to play either standing or sitting [25, 29]. Finally, the German Gymnastics Association aimed to reduce the dropout rate of people with dementia, allowing those affected to participate in their sports groups for as long as possible. Thus, exercise instructors received educational training to be sensitized to the clinical picture and be able to work with those affected by recognizing their specific needs within the exercise session while keeping them integrated and active in their sports group [25, 30].

## Sample and recruitment

This qualitative descriptive study was conducted with project leaders (PLs) and sports trainers (STs) from the four sports associations involved in the project *"Sport bewegt Menschen mit Demenz" (Sports moves people with dementia)*. PLs were higher-level administrators within the associations managing and coordinating the sports initiatives. Additionally, the majority of STs were qualified, B-licensed exercise instructors focused on health, prevention and rehabilitation. Recruitment of PLs and STs was carried out through purposively sampling for recruiting participants with the best chance of obtaining relevant and meaningful data [31]. The DOSB project coordinator was the person in charge of help finding suitable interview participants. Thus, PLs and STs from the German Sports Associations of Lower Saxony, the German Sports Associations of North Rhine-Westphalia, the German Table Tennis Association, and the German Gymnastics Association, participating in the project, were contacted and invited via email to be interviewed. Thus, each PLs and STs received a personal invitation to be part of a semi-structured interview. Every person reached was interested in being part of the study.

## Implementation science framework

We used the CFIR as a guiding framework for data collection and analysis as it provides a comprehensive, holistic and dynamic series of different constructs grouped into five domains that have been linked with successful implementation across multiple levels [32]. Thus, the CFIR framework offered a social-ecological perspective and an ideal multilevel structure for providing evidence on barriers and facilitators that need to be addressed if PA interventions among PwD are to be successfully implemented [33]. Table 1 describes CFIR domains included in this study.

## Data collection

Qualitative data was collected through semi-structured interviews. Thus, interviewers followed a qualitative semi-structured interview guide designed based on a template of CFIR constructs accessible at https://cfirguide.org/tools/ (see S1 Table). This template allowed us to select twenty constructs and questions relevant to the study [32]. Also, we could adjust the guide and include other questions relevant to the project. Finally, members of DOSB and DALzG reviewed this guide to ensure its quality and relevance. Due to the COVID pandemic restrictions, semi-structured interviews were conducted via videoconference using the Zoom platform between December 2021 and February 2022. A total of 13 interviews were conducted by trained psychologist with experience in qualitative research. In total 21 participants, PLs (n = 6) and STs (n = 15) completed the interviews. The average duration of each interview was one hour and forty-five minutes. All interviews were recorded and transcribed verbatim. The interviews were conducted in German and translated into English after transcription. All

**Table 1. Definitions of the CFIR domains and constructs included in this study [33].**

| CFIR Domain | CFIR Construct | Definition |
|---|---|---|
| **Intervention Characteristics** | Relative advantage | The perceived benefit of executing the intervention over a different strategy |
| | Adaptability | The extent that the intervention is adequate to meet the specific local needs |
| | Intervention Source | The perception of whether the intervention development originates from external or internal sources |
| | Complexity | Perceived difficulty of implementation |
| **characteristics of individuals** | Motivation | The stage an individual has reached as he or she moves toward becoming a skillful and enthusiastic adopter of the intervention |
| | Knowledge and beliefs about the intervention | Attitudes towards the value of the intervention |
| | Self-efficacy | Individuals' belief in their own abilities to carry out certain actions |
| **Process** | Planning | The degree of development of an anticipated concept and method regarding the execution tasks |
| | Formally appointed internal implementation leaders | Individuals formally designated inside the organization with the responsibility of delivering the intervention |
| | Engaging | Involvement of relevant individuals for the intervention using a social marketing strategy, education, training, etc. |
| | Executing | Execute implementation as planned |
| | Reflecting and evaluating | Collection of quantitative and qualitative information on the advances and quality of project implementation |
| **Inner setting** | Implementation climate | The degree of the organization's capacity for change and receptiveness towards the use of an intervention |
| | compatibility | The extent to which the significance and attributed values of the intervention are in line with current work practices |
| | Learning climate | The environment allows individuals to express their need for assistance, to feel appreciated and confident to try new methods, and to find time for discussion and evaluation |
| | Relative Priority | Shared perception among individuals concerning the implementation's relevance |
| | Communication | The type and quality of formal and informal communication that exists within the organization |
| | Available resources | Resources allocated for implementation, such as funds, training, physical space, and time |
| **Outer Setting** | Cosmopolitanism | The extent to which an organization is connected to external organizations |
| | External policy and incentives | Contemplates external strategies to disseminate actions and interventions, such as policies, regulations, guidelines, etc. |

transcriptions were also pseudonymised. The illustrative quotations were translated into English for the present publication and proofread by a native German speaker.

## Data analysis

Conventional qualitative content analysis (QCA) was applied to provide descriptive insights and an understanding of the understudied subject [34, 35]. The textual data (the transcripts) was read and then text fragments were analyzed to be associated with relevant codes. When developing the coding framework, we used a unified deductive and inductive category approach. In the case of the deductive categories, they were drawn out of the semi-structured interview guide based on the CFIR domains and constructs, while the inductive subcategories arose at the time of the interview. Thus, interviews were used to explore further subcategories to identify areas relevant to a broader scope of discussion. To ensure the coding framework was comprehensive, interviews were thoroughly reviewed multiple times. In cases in which no other categories could be extracted from the interview transcripts, a line-by-line analysis of all the interviews was carried out using the coding framework [34, 35]. Additionally, through the interviews, we quantified the total number of barriers and facilitators encountered when implementing the project according to the perceptions of the 21 participants.

## Results

Main results obtained from the interviews describing facilitators and barriers for PA initiatives implementation among PwD can be categorized into the five main CFIR domains: 1) Intervention Characteristics, 2) characteristics of individuals, 3) Process of implementation, 4) Internal setting, and 5) Outer setting. Table 2 shows the number of facilitators and barriers identified according to participant's perceptions, based on the CFIR framework.

**Table 2. Barriers and facilitators for PA initiative implementation based on the CFIR framework.**

| CFIR Domain | CFIR Construct | Number of barriers and facilitators identified according to participants perceptions (n = 21) | | | Quotation |
|---|---|---|---|---|---|
| | | Barrier, yes | Neutral or not specified | Facilitator, yes | |
| **Intervention Characteristics** | Relative Advantage | 0 | 0 | 21 | "Sports offerings for older people [. . .] is a very important point here. The clubs that expand their offerings to include such activities are also positioned for the future from our point of view [. . .]. And of course, the topic of inclusion, i.e. of people with illnesses, with disabilities, is also a relevant field of action". PL3 |
| | Adaptability | 4 | 8 | 9 | "The project is certainly necessary when you see that there will be more and more people who have dementia. People are getting older and older and you are more likely to meet people with dementia". ST2 |
| | Intervention Source | 0 | 3 | 18 | "The advantage, of course, is always that when you have something like the DOSB and the DAlzG, some large sponsorship behind it, you often don't have to worry about how it's going to be financed. It has to be a very big factor, especially for an association". ST10 |
| | Complexity | 19 | 2 | 0 | "The obstacles lie in the persons themselves or in the relatives and 'I don't want to bring or pick up my relatives, I don't have the time' or they say 'No, I can't walk' or 'I'm not going to start doing sports in my old age'. ST13 |
| **Characteristics of Individuals** | Motivation | 0 | 5 | 16 | "PA helps PwD health, well-being and so on. And knowing that there is relatively little offered for dementia patients, it is my incentive to say, hey, I'm going to help these people, offer something for them to get them out of their home once a week, have exercise, have joy and fun". ST10 |
| | Knowledge and Beliefs about the Intervention | 3 | 5 | 13 | "I think the project is very important because sports really changes the development process of the disease, because people have more fun, because it brings inner". ST7 |
| | Self-efficacy | 0 | 5 | 16 | "So, on a scale of one to five on how confident I feel implementing the project, I'm a four since we implemented the project. I won't say I'm a five since there is the issue of Corona" PL1 |
| **Process** | Planning | 5 | 2 | 14 | "Yes, we have a method, we have created a project concept and I have also created a rough schedule with a milestone plan". PL6 |
| | Engaging | 0 | 0 | 21 | "I personally find the fact that we do this together with the project management a very great added value, because we both have our own areas of expertise and I find that in this case they very well matched.". PL4 |
| | Executing | 0 | 0 | 21 | "We have implemented the project always trying to address as many different aspects as possible. We always started with memory training and social interaction [. . .]. Then we did muscle training combined with balance exercises [. . .] we tried to include some kind of dual tasting exercises. ".ST10 |
| | Reflecting and Evaluating | 14 | 2 | 5 | "No, we didn't do any evaluations, unfortunately. So, we talked briefly with the supervisors afterwards and exchanged ideas about what they saw, what improvements, how they received it, or how the sports program was going. But that is not documented". ST4 |

*(Continued)*

**Table 2.** (Continued)

| CFIR Domain | CFIR Construct | Number of barriers and facilitators identified according to participants perceptions (n = 21) | | | Quotation |
|---|---|---|---|---|---|
| | | Barrier, yes | Neutral or not specified | Facilitator, yes | |
| **Inner Setting** | Implementation Climate | 3 | 4 | 14 | "I would say the association is committed in the sense of wanting to participate and provide support". ST1 |
| | Compatibility | 0 | 3 | 18 | "We are open to the project because it just fits perfectly so far in our strategy". PL2 |
| | Learning Climate | 0 | 6 | 15 | "There were training offers that actually supported us a lot in our project progress". PL6 |
| | Relative Priority | 10 | 6 | 5 | "So given the other issues that are on the agenda right now, it's certainly not the highest priority". PL6. |
| | Communication | 6 | 4 | 11 | "This exchange between STs and supervisors doesn't take place regularly or at all". ST6. |
| | Available Resources | 1 | 5 | 13 | "Financial resources are provided to us through project funds. That is an essential factor to carry out the project".PL1 |
| **Outer Setting** | Cosmopolitanism | 18 | 0 | 3 | "In our case, there is not necessarily good networking between the individuals, care facilities and so on, i.e. doctors, PwD, sports clubs. If there was a better network, i.e. better cooperation overall, then I think it would be easier to reach more people". ST4 |
| | External Policy and Incentives | 0 | 19 | 2 | "Regulations and guidelines have influenced the decision to implement the measures and above all these whole project". ST9 |

## Theme 1: Intervention characteristics

The following intervention characteristics facilitated the execution of this sports project among PwD. All participants (n = 21) believed there was an *advantage* of implementing this project within their sports associations and almost the half (n = 9) perceived that this sports project was *adaptable* to fill existing gaps at the local level. In addition, most of them (n = 18) perceived positively the *intervention source* and the external institution that developed it (e.g., DOSB, DAlzG). They felt supported by them in providing funding, training and contacts with other experts.

> *"We've been dealing with this topic for at least ten years, but we get more and more feedback from exercise instructors saying that memory problems are becoming more evident in some of the long-term club members. Thus, exercise instructors ask "What can we do? How do we deal with this? And we are already seeing quite a lot of interest and need for information about the disease. And from there, this project could be an important step forward to address these questions". PL6*

In contrast, the majority of participants (n = 19) expressed concerns on the *complexity* of the project. Most participants felt that the innovation was complex due to low participation rates associated with stigma towards dementia, the pandemic, and characteristics of the target group (e.g. stage of dementia, dementia symptoms).

> "*You have to bring a lot of energy in order to place this topic at all. It is still stigmatized, I would think. And it's not easy for the associations to communicate that they have appropriate offers and to reach participants. So, these are major obstacles, reaching people and it is certainly the case that it is still a negative and fearful topic*". PL6

## Theme 2: Characteristics of individuals

The characteristics of individuals that could influence the success of project implementation were associated with majority of PLs and STs (n = 16) being highly *motivated*. Also, most of them (n = 13) expressed *positive opinions about the innovation* and *felt confident* (n = 16) about their own knowledge and skills to deliver the interventions.

> *"I feel very confident, thanks to the trainings I have received and my experience in three other projects". ST9*

However, the pandemic restrictions and the low number of participants led them (n = 5) to some uncertainty.

> *"We were always sure that the project could be carried out and that it was good and important. However, the only factors that made it so uncertain for us were the pandemic and the number of participants". ST11*

## Theme 3: Process

As a facilitator factor in the process domain, it was found that, according to 16 participants, most associations incorporated a detailed *plan* with the necessary steps for implementation. Besides, the 21 participants affirmed that all associations formally appointed internal implementation leaders for executing the sports initiatives and all *accomplished the implementation* of the project.

> *"Yes, we have a method. We have created a project concept and I have also created a rough schedule with a milestone plan". PL6*

We identified as a barrier that according to 14 participants there was no quantitative or *qualitative assessment* on the progress of implementation.

> *"The limited funds available do not allow for an external evaluation. Also, an external evaluation, or even an internal evaluation, exceeds the possibilities of a non-profit association both financially and in terms of manpower". PL5*

## Theme 4: Inner setting

Some characteristics of the associations and sports clubs facilitated the implementation of the sports initiatives. The majority of participants (n = 14) agreed that the project has a positive implementation climate. Moreover, most of them (n = 18) mentioned that the project was *compatible* and in line with existing work processes, norms and values, and the needs of association members. In addition, most participants (n = 15) mentioned a positive *learning climate* where associations offered training opportunities on the topic sports and dementia.

> *"I would say that the association and the board are very open to new offers. So they tend to be immediately involved when it comes to such new initiatives, especially in the area of older people". ST4*

On the contrary, almost the half (n = 10) of PLs and STs perceived some barriers within the associations and sports clubs in terms of a low *priority* towards dementia and sport compared

to other topics. Also, major part of participants (n = 11) mentioned that even though communication channels and working groups existed within their associations, there was still a need to increase the possibility of more *exchange* between STs and board members. Finally, large number of participants (n = 13) were concerned about the available resources as they were not sure about how to continue obtaining financial support in order to maintain the sports proposals in the long term.

> *"I think that the offers are already connected with an enormous effort for the exercise leaders. So we need funding to continue counting on them".* PL6

### Theme 5: Outer setting

A few participants (n = 2) mentioned that *external policies and incentives*, including national strategies, policies, recommendations, and guidelines for PwD, facilitated the project implementation as those promote action and funding of such projects.

> *"The federal government has really decided something good, I must say, with the National Dementia Strategy, there is an impulse on how to help older demented people".* ST1

On the other hand, the majority (n = 18) of participants mentioned that *networking with external structures* is still needed to propose sustainable initiatives. According to participants, it is possible to overcome financing, training, or attendance gaps through networking.

> *"I always depend on cooperation and collaborations. I could not achieve so much on my own. That means we need the support and cooperation of the clubs. Also, the municipal structures are actually essential for us. This is where people live, where organized sports are networked with the professional societies in the community and where they have to present their services".* PL6

## Discussion

### Main findings

In this research, we applied an implementation of science framework, the CFIR, to provide a more profound and multilevel insight into the process of implementing sports initiatives for PwD. Twenty factors (barriers and facilitators) defined by the CFIR framework affected the DOSB project's delivery, influencing initiative successful. Through the description of these factors, we want to offer a guide towards decisive determinants that potentially enlighten future sports initiatives implementation in this population.

**Intervention level factors.** Different factors played a fundamental role at this level and served as facilitators of the sports initiative, including *relative advantage, adaptability, and intervention source*. According to the literature, when the interested parties consider that the innovation's implementation has an unquestionable benefit and advantage in effectiveness, its application will be more likely to be successful [20]. Moreover, this knowledge about advantages contributes to positive attitudes towards initiatives that influence the effort to demonstrate the initiative's benefits, which will also help its implementation [36]. Therefore, *relative advantage* is an indispensable premise for an innovation to be accepted and implemented [37]. In line with the literature, our study participants showed a perceived relative advantage from the project as they manifested that offering sports initiatives for PwD within their clubs gave

them a higher positioning than other sports clubs by offering more courses and inclusive programs. Additionally, participants stated that the project was very beneficial for PwD, as sports opportunities for this population are scarce but necessary due to their positive impact on physical, mental and social health. This perception of the projects' advantages impacted practice because participants made a great effort to compromise with the project and demonstrate its efficacy. The practical application of these results might be translated into future sports initiatives involving educational strategies among PA providers and their settings that promote knowledge about benefits and advantages of PA in PwD. This may raise dementia awareness and future efforts to make sporting opportunities available to this population.

*Adaptability* was also an important implementation facilitator factor. A project that can easily be adjusted to address specific community needs has a higher chance of successful implementation [33, 38]. Results of this study support the previous statement as participants recognized that the project was easily modified to meet specific local need. For example, participants indicated that the project was adapted to the needs expressed by the association trainers, who stated that memory problems were becoming more and more evident in the long-term members. However, they needed more knowledge and tools to work with these members. Thus, their initiative within the project offered training courses for STs on dementia and sports to address lack of information manifested by STs. Moreover, they recognized that training sessions were a significant step forward and might have influenced the success of the project implementation as STs felt more self-confident and empowered to work with the target population. In this way, adaptability should be considered in practice when developing and applying PA initiatives. Those responsible for developing proposals should know their context's needs and define core components (elements that cannot be changed) to ensure the implementation is relevant to their target population and adaptable periphery components (elements that can be changed of the innovation) to allow flexibility and adaptation for a variety of contexts [39].

*Intervention source* also stood out as a facilitating factor for the interviewees. In line with previous literature, a positive perception, whether the project is externally or internally developed, play an essential role in influencing effective implementation [33, 40]. Our results prove that study participants positively perceived that the project originated from external organizations (e.g. DOSB, DAlzG). They saw them as reputable, credible and trustable organizations which encouraged participation among actors to make decision and supported implementation providing from the beginning expertise, solutions, and previous efforts that served as a solid base for the subsequent execution of the project within the sports associations. In such a manner, involved actors from future PA strategies must perceive that change is not imposed on them by intervention sources; on the contrary, actors should feel included in the decision-making process when building change [40]. Those agents who impose change with low participation of those concerned are more likely to be ineffective in implementation [41]. Therefore, the success of the implementation will depend on the trust generated based on the participation between the source of the intervention and the participants.

In contrast, the *complexity* of the initiative was perceived as a relevant barrier influencing the initiative's success. Complexity is defined by multiple interacting elements that often bring out low participation rates, variability in context, content, mode of application and uncertainty of the impact [42]. According to the CFIR framework, the perceived complexity of implementation at the interventional level also influences the effective execution of the project [32]. Our results show that participants expressed high complexity in developing sports initiatives aimed at PwD. These difficulties interfered with project implementation, affecting, in most cases, the number of PwD involved. Mainly, the complexity of the intervention, affecting low participation rates, resulted from three components: 1) Behavioral components and actions on behalf of the target population. For instance, PwD evidenced fearful behaviors when revealing their

diagnosis to others. Especially people in the early stages of dementia were afraid to admit their diagnosis when participating in the sports initiative designed for PwD. Consequently, they preferred not to take part in these kinds of initiatives. Moreover, low motivation levels were also noticed among PwD as they manifested beliefs such as "working out requires more effort than the benefits it brings; I am too old to start exercising". These results are consistent with findings from previous studies, showing that loss of motivation in PA is present in at least 70% of PwD [43, 44]. Likewise, low willingness was noticed from caregivers to bring and accompany persons with dementia in the sports sessions; 2) Community components, where sports associations manifested that dementia in their sports clubs was a fearful topic as STs and members lacked awareness and had a negative attitude against the disease; And 3) Contextual component, characterized by the COVID-19 pandemic, which interfered with the PA project implementation and participation of PwD as STs were forced to change their initial implementation ideas, adapt and restructure their plans according to the new restrictions and provide activities and spaces where it was safe to practice sports. Thus, those insights into the intervention's complexity may be helpful for the successful implementation of future initiatives. The ability to illustrate and reproduce intervention's complexity matters, given their extensive application across health and social support services and the increased concern for improving our understanding of their efficacy and outcomes [45]. Eventually, to illustrate project's complexity, might lead to an in-depth comprehension of their action processes and impacts, i.e., how and why they operate in a particular setting [46]. In this particular project, the perceived complexity at the intervention level demonstrates that there is still a great need to sensitize and educate sports clubs and communities about the disease. Also, it is necessary to target motivational levels, including intra- and interpersonal aspects, when developing sports initiatives for PwD, such as preferred and enjoyable PA offers, promoting positive experiences and attitudes towards PA, and providing social support [13, 47]. Finally, the pandemic highlighted the importance of having PLs and STs responsible for PA initiatives with the resources and time to adjust plans and develop new ideas according to the inconveniences that might arise during implementation.

**Individual level factors.** At the individual level, three main facilitators were found for successful sports initiative implementation: motivation, knowledge and beliefs about the intervention and self-efficacy. According to the CFIR framework, the individual level emphasizes the significance of people's roles and features in the innovation's execution and delivery, mainly since individuals' actions and behaviors drive the project's implementation and personal attributes also mediate the effective delivery of the project [32]. This study supports this statement as *highly motivated and self-confident* PLs and STs implemented the project despite the various challenges that such an initiative entails. Thus, care practice and future PA initiatives for PwD should capture and incorporate strategies to enhance PA provider's motivation, knowledge and beliefs about the intervention and self-efficacy as determinants for successful implementation, as mentioned in the literature [48]. Motivation understood as the willingness of individuals to undertake an action because they are intellectually and emotionally connected to it [49]. Accordingly, when individuals are motivated, such enthusiasm can translate into the effective use of innovations as positive subjective opinions, personal experiences, perceptions, and attitudes towards a new behavior precede engagement and change [50, 51]. In addition, self-efficacy, individuals' belief in their ability to carry out roles and make a change [52], is a relevant factor for the successful implementation of the initiatives because those who feel confident in their skills tend to be more motivated and committed to participating and implementing the proposals despite obstacles [20]. One helpful way to understand and improve the willingness and readiness of people involved in implementing initiatives is the Stages of Change Model (Transtheoretical Model), as it outlines the way an individual builds up new

behaviors through different stages (pre-contemplation, contemplation, preparation, and action and maintenance) [53]. Different strategies can support individuals to move forward into the next phase and thus gradually move toward skillful, enthusiastic, confident and motivated PA providers who are needed to implement successful innovations [20].

**Process level factors.**    At the process levels, different determinants also played a fundamental role for successful implementation, including planning, formally appointed internal implementation leaders, engaging and execution. Different theories from the perspective of quality management and integrated care provide a set of principles that illustrate how implementations should be conducted to be successful, thus, most approaches agree on four activities: planning, engagement, implementation and evaluation [33]. Consistent with the literature, this study supports these theories as *planning, engagement and implementation* were used to implement the initiatives successfully. Nevertheless, it was not possible to implement an *evaluation* because evaluation processes exceeded the possibilities of non-profit associations in terms of staffing and lack of funding, which underlines that in future, such projects should focus on ensuring that there are enough resources to support projects at every stage. In this way, future programs could consider such implementation activities (planning, doing, evaluating) so that their projects have structures and processes to ensure sustained effort and change.

**Inner setting level factors.**    At the *inner setting level*, implementation climate, compatibility, learning climate, and available resources were facilitator factors for implementation. The CFIR framework states that the degree to which an intervention's application is valued, supported, and expected within an organization impacts its success [32]. This is reflected in our results since the sports associations showed compatibility (with current work practices), had a positive *implementation climate* (receptiveness towards the initiative), provided a *positive learning climate*, and provided *resources* facilitating the project implementation. In particular, most participants indicated that it was essential for implementation that the project provided associations with resources, including financial support for covering costs associated with the payment of STs, the provision of materials, and the availability of educational opportunities. On the contrary, *relative priority* and *communication* were factors considered by participants as barriers for implementation. In this way, the inner setting should also be addressed within the practice as a factor that has an active role in interacting with the implementation. As such, future initiatives could include an objective description of their settings, i.e., specialization, size, maturity, and age, as these may be positively or negatively associated with implementation [54].

**Outer setting level factors.**    Finally, at the *outer setting level*, *cosmopolitanism* and *external policy* and incentives were influencing implementation factors. According to the literature, organizations that are supported and encouraged by external agents are more likely to implement new practices readily and sustain them over time [32]. This was reflected in the results of our assessment, as the participants stated that having a solid network, the support of DOSB and DAlzG, and active national policies allowed them access to the necessary resources to carry out the project. However, the participants remained concerned about the need for a more robust networks with other associations and relevant actors in this field (policymakers, PLs, STs, PwD, general practitioners, neurological clinics) for sustainable proposals as networking encourages active participation, exchange of knowledge, and funding opportunities [33]. Thus, for further sports initiatives to be implemented, should take into account macro-level factors, as connections and changes in the outer setting can positively or negatively impact the implementation.

**Strengths and limitations.**    Planning and delivering sports projects for PwD is challenging, as different interpersonal, organizational, and community factors influence the success of these initiatives. Therefore, this study corroborates the potential of the CFIR as a conceptual

guide for the successful implementation of sports projects in PwD, mainly since it permits the identification of multilevel factors that will affect large-scale project application. Moreover, in this study, we have focused our attention on sport association and their sport clubs, contexts that play a crucial role in persons PA engagement in Germany because of the ongoing sport opportunities they provide to a large sector of the population. Therefore, we believe that having qualitative data from this precise context, provides important insights about factors and barriers that might support the development of future strategies to increase PA participation among PwD, optimize the benefits of PA and prolong the sustainability of such initiatives. When interpreting the results of the present study, account should be taken on the following limitation. While we sought to collect data on factors affecting the implementation of sports initiatives at different levels, we interviewed PLs and STs from the sports associations. Therefore, the interview responses and our findings represent the individual perspectives shared by PLs and STs, rather than the perspectives of PwD and their caregivers.

## Conclusions

This study the CFIR allowed recognizing facilitator factors to successful implementation particularly at the individual, process and inner setting level. Participants were highly motivated, showed positive self-efficacy and attitudes towards the project. Moreover, the associations provided detailed planning, appointed leaders, and resources needed for implementation. In addition, the CFIR helped detect barriers that require attention to successfully implement sports initiatives long-term, particularly at the intervention and outer setting levels. For example, more financial incentives are needed to encourage and support STs working with and for PwD long term. Also, there is the need to create solid networks for sustainable proposals making visible platforms that enable knowledge exchange between interested parties. This study also evidenced the lack of evaluation processes to assess the effectiveness of the initiatives, not only evaluation processes to assess whether the objectives had been met, but also did not measure other relevant outcomes that could account for the effectiveness of the initiatives, such as changes in PA or psychosocial factors after participation in this far-reaching project. It is essential that future projects evaluate their effectiveness also, including those measurement outcomes, to maximize benefits and assess the true scope of such projects. Scientific research has much to contribute in this sense; therefore, future synergies between practice and science are still needed; both can benefit from this collaboration. Finally further research is invited to consider conceptual frameworks emphasizing multilevel ecological factors as they give Indicators to carry out integral sport approaches to cover relevant implementation areas needed to promote PA participation in PwD.

## Supporting information

**S1 Table. Semi-structure interview guide.**
(PDF)

## Acknowledgments

Gisela Nüssler, the DOSB project coordinator, who facilitated this evaluation by mediating between the researchers and the DOSB board. She also provided the contact information of participants and ensured that the interview guides were aligned and relevant to the project.

## Author Contributions

**Conceptualization:** Maria Isabel Cardona, Jochen René Thyrian.

**Methodology:** Maria Isabel Cardona, Jessica Monsees, Tim Schmachtenberg, Anna Grünewald.

**Supervision:** Jochen René Thyrian.

**Writing – original draft:** Maria Isabel Cardona.

**Writing – review & editing:** Maria Isabel Cardona, Jessica Monsees, Anna Grünewald, Jochen René Thyrian.

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
