## [Decision Letter · Decision Letter 0]

15 Mar 2023

PONE-D-22-26436Implementing a physical activity project for people with dementia – identification of barriers and facilitator using consolidated framework for implementation research (CFIR): A qualitative studyPLOS ONE

Dear Dr. Cardona,

Thank you for submitting your manuscript to PLOS ONE. After careful consideration, we feel that it has merit but does not fully meet PLOS ONE’s publication criteria as it currently stands. Therefore, we invite you to submit a revised version of the manuscript that addresses the points raised during the review process.

We look forward to receiving your revised manuscript.

Kind regards,

Eshak I Bahbah

Academic Editor

PLOS ONE

Journal Requirements:

Reviewers' comments:

Reviewer's Responses to Questions

**Comments to the Author**

1. Is the manuscript technically sound, and do the data support the conclusions?

Reviewer #1: Yes

Reviewer #2: Yes

Reviewer #3: No

Reviewer #4: Partly

2. Has the statistical analysis been performed appropriately and rigorously? 

Reviewer #1: Yes

Reviewer #2: N/A

Reviewer #3: No

Reviewer #4: No

3. Have the authors made all data underlying the findings in their manuscript fully available?

Reviewer #1: Yes

Reviewer #2: Yes

Reviewer #3: No

Reviewer #4: No

4. Is the manuscript presented in an intelligible fashion and written in standard English?

Reviewer #1: Yes

Reviewer #2: Yes

Reviewer #3: Yes

Reviewer #4: Yes

5. Review Comments to the Author

Reviewer #1: This is an interesting study that aimed to identify multilevel barriers and facilitator factors when implementing a project designed to promote physical activity participation for people with dementia in Germany. Some suggestions are listed below for the authors to consider for improvement of the paper.

1. Since this study was conducted in Germany, please indicate in the manuscript title.

2. Please talk about the knowledge gaps clearly. For example, what factors that have not been investigated by previous studies based on the CFIR domains or constructs.

3. The authors should give justifications for using the CFIR framework in this study. Why did the authors choose the CFIR, but not others (e.g., the Theoretical Domains Framework). One of the major advantages of using the CFIR is that this framework has been widely applied. There are a few systematic reviews that have used the CFIR to investigate facilitators or barriers to implementing a health intervention. Please cite such publications when talking about the reasons for using the CFIR. One example is listed below. Please cite where appropriate.

Chan, P.Sf., Fang, Y., Wong, M.Cs. et al. Using Consolidated Framework for Implementation Research to investigate facilitators and barriers of implementing alcohol screening and brief intervention among primary care health professionals: a systematic review. Implementation Sci 16, 99 (2021). https://doi.org/10.1186/s13012-021-01170-8

4. For data analysis, this is a very important part. The authors should give details how the data coding of the data was performed. Any detailed approaches or procedures employed? Since the definitions of the CFIR constructs would be very different from the determinants of the study results, how were they matched? Given the current form, it is not clear about the data coding process.

5. It is expected that there should have some discussions about implementation strategies that address the determinants in order to improve physical activity among people with dementia. There are a host of implementation strategies. The authors should consider adding this section to the paper.

Look forward to reading the revised version of this interesting paper.

Reviewer #2: I have read and assessed the manuscript. I think the author studied a good topic about dementia patients which is a qualitative one with no needed statistical analysis. Concerning the language, it is acceptable with sound grammar and context. I think the author was able to include more interviews and participants, but the 20 is good.

Reviewer #3: As a reviewer, I cannot recommend rejecting the paper solely based on the methods and results presented. However, I have some major concerns.

Firstly, the study's sample size is small and may not provide a comprehensive view of the subject under study. The study only included project leaders and sport trainers from four German sports associations, which could lead to sampling bias.

Secondly, the study's data collection and analysis are unclear. The study only stated that semi-structured interviews were conducted via video conference using Zoom and that the interviews were recorded and transcribed verbatim. However, the study did not mention how the interviews were analyzed or provide any specific coding methods.

Finally, the study's results are not presented in a clear and organized manner. The study presents the results by CFIR domains, but the presentation is not comprehensive and may confuse the reader. It is unclear how the negative, neutral, and positive perceptions were measured or how they contribute to the results.

Therefore, I recommend rejecting the paper due to the ethical concerns and the lack of clarity in the methods, data collection, and analysis.

Reviewer #4: First of all, the authors said "ethical approval from a committee was not sought" and this research included a patients, even there was no interventions, this kind of research also require ethical approval.

How the authors did not include participants that were patients? Dose the PwD not patients???

"After study approval, members of the steering committee informed project staff about our study, and those interested in participating could do it voluntarily." approval from who? the authors said firstly that they did not sought ethical approval, then they said after study approval. ?????

6. PLOS authors have the option to publish the peer review history of their article (what does this mean?). If published, this will include your full peer review and any attached files.

Reviewer #1: No

Reviewer #2: **Yes: **Ahmed Aljabali

Reviewer #3: No

Reviewer #4: No

<quillbot-extension-portal></quillbot-extension-portal>

---

## [Author Response · Author response to Decision Letter 0]

29 Apr 2023

Response to Reviewer # 1 

Thank you for your review of our paper. Please find our point-to point reply below.

Point 1: Since this study was conducted in Germany, please indicate in the manuscript title.

Response 1: We agreed with the reviewer on this important remark. Therefore, we modified the tittle as follows: 

Line 1: “Implementing a physical activity project for people with dementia in Germany – identification of barriers and facilitator using consolidated framework for implementation research (CFIR): A qualitative study”.

Point 2: Please talk about the knowledge gaps clearly. For example, what factors that have not been investigated by previous studies based on the CFIR domains or constructs. 

Response 2: We thank the reviewer for this important comment. As a result, changes were made in the introduction to clarify what knowledge gaps are evident in the existing literature. We emphasised on the fact that although many studies have focused on identifying barriers and facilitators mediating physical activity outcomes, previous studies do not use scientific frameworks for implementation. Which outline in-depth barriers and facilitators at different levels that influence the success of interventions.

Line 93: “In spite of the existing evidence on the potential barriers and facilitators mediating PA outcomes in PwD, prior research did not provide a scientific angle using frameworks for implementation research. According to the most recent systematic reviews published in 2016 and 2018, studies synthesized barriers and facilitators to PA adherence in PwD, mainly including qualitative methodologies using data collection strategies like interviews and focus groups. However, to narrow the gap from research findings to practice, a comprehensive insight into the barriers and facilitators is required to bring more effective implementation strategies”.

Line 116: “Regardless of the existence of such frameworks, their value to the field of dementia care and PA remains limited. Therefore, there is a need for further research that utilizes implementation science frameworks to outline in-depth barriers and facilitators of the intended change in practice at different levels to clarify the potential drivers of change and the challenging implementation dynamics and enable improved dementia care PA initiatives outcomes”. 

Point 3: The authors should give justifications for using the CFIR framework in this study. Why did the authors choose the CFIR, but not others (e.g., the Theoretical Domains Framework). One of the major advantages of using the CFIR is that this framework has been widely applied. There are a few systematic reviews that have used the CFIR to investigate facilitators or barriers to implementing a health intervention. Please cite such publications when talking about the reasons for using the CFIR. One example is listed below. Please cite where appropriate.

Chan, P.Sf., Fang, Y., Wong, M.Cs. et al. Using Consolidated Framework for Implementation Research to investigate facilitators and barriers of implementing alcohol screening and brief intervention among primary care health professionals: a systematic review. Implementation Sci 16, 99 (2021). https://doi.org/10.1186/s13012-021-01170-8

Response 3: Many thanks for the valuable remark. The introduction was modified to justify why we chose the CFIR framework for the study. Moreover, we cited two more systematic reviews, in addition to the one suggested, to support the fact that this framework has been widely used.

Line 105: “One widely applied science framework is the Consolidated Framework for Implementation Research (CFIR) [21-23], which is used as a guiding tool for the multilevel assessment of implementation settings to detect facilitators and barriers to the successful implementation of interventions [22, 23]. Furthermore, it offers a set of standardized and comprehensive implementation constructs, which account for characteristics that may influence implementation and thus play an essential role in whether or not interventions are successful [22, 23]. Thus, this model groups these constructs into five domains: (1) Intervention characteristics (characteristics of the initiative likely to affect implementation); (2) Inner setting (organizational characteristics likely to affect implementation); (3) Outer setting (environmental characteristics likely to affect implementation); (4) Characteristics of individuals (persons engaged in the initiative likely to affect implementation); and (5) Implementation process (planning, executing, evaluating the intervention) [22, 23]”. 

Point 4: For data analysis, this is a very important part. The authors should give details how the data coding of the data was performed. Any detailed approaches or procedures employed? Since the definitions of the CFIR constructs would be very different from the determinants of the study results, how were they matched? Given the current form, it is not clear about the data coding process.

Response 4: Thank you for this remark. Again, it has been of great help in improving the clarity of the manuscript. Therefore, we have modified the text, adding a better description of how the data coding was performed, as follows:

Line 241: “When developing the coding framework, we used a unified deductive and inductive category approach. In the case of the deductive categories, they were drawn out of the semi-structured interview guide based on the CFIR domains and constructs, while the inductive subcategories arose at the time of the interview. Thus, interviews were used to explore further subcategories to identify areas relevant to a broader scope of discussion. To ensure the coding framework was comprehensive, interviews were thoroughly reviewed multiple times. In cases in which no other categories could be extracted from the interview transcripts, a line-by-line analysis of all the interviews was carried out using the coding framework”. 

Point 5: It is expected that there should have some discussions about implementation strategies that address the determinants in order to improve physical activity among people with dementia. There are a host of implementation strategies. The authors should consider adding this section to the paper.

Response 5: 

Thank you for the suggestion. We agree that it is crucial to consider some discussions about implementation strategies to address determinants and improve physical activity in people with dementia. However, in this paper, we decided to focus more on the implementation process and offer guidance on those determinants that might be needed to illuminate future interventions. Mainly since very few studies concentrate on methods and processes to achieve change in strategies for people with dementia.

Response to Reviewer # 2 

Point 1: I have read and assessed the manuscript. I think the author studied a good topic about dementia patients which is a qualitative one with no needed statistical analysis. Concerning the language, it is acceptable with sound grammar and context. I think the author was able to include more interviews and participants, but the 20 is good.

Response 1: 

Point 1: We appreciate the reviewers' time in revising the manuscript. We are sincerely thankful for their valuable comments and observations, which have greatly contributed to raising this manuscript's perceived quality. 

Response to Reviewer # 3 

Point 1: Firstly, the study's sample size is small and may not provide a comprehensive view of the subject under study. The study only included project leaders and sport trainers from four German sports associations, which could lead to sampling bias.

Response 1: we would like to thank you for your review and comments. As you mentioned, sampling is a crucial aspect of providing comprehensive results. Thus, we took seriously the sample size of this research. Since this is qualitative research, methodological studies in this field have shown that sample sizes between 17 and 40 participants achieve data saturation of meta-themes, guaranteeing rigorous quality (1-3). Furthermore, small sample sizes underpin detailed, in-depth case analysis, which is vital in this research approach (4). Accordingly, we believe that a sample of 21 participants is in line with the literature and enables a comprehensive view of the subject of the study.

1. Hagaman AK, Wutich A. How many interviews are enough to identify metathemes in multisited and cross-cultural research? Another perspective on guest, bunce, and Johnson’s (2006) landmark study. Field Methods. 2017;29(1):23–41.

2. Francis JJ, Johnston M, Robertson C, Glidewell L, Entwistle V, Eccles MP, et al. What is an adequate sample size? Operationalising data saturation for theory-based interview studies. Psychol Health. 2010;25(10):1229–45.

3. Vasileiou, K., Barnett, J., Thorpe, S. et al. Characterising and justifying sample size sufficiency in interview-based studies: systematic analysis of qualitative health research over a 15-year period. BMC Med Res Methodol 18, 148 (2018). https://doi.org/10.1186/s12874-018-0594-7

4. Sandelowski M. One is the liveliest number: the case orientation of qualitative research. Res Nurs Health. 1996;19(6):525–9

Additionally, we would also like to acknowledge that a limitation of our study is that we did not include people with dementia, which would have enriched our results. However, only project leaders and trainers were included in our study since we aimed to identify determinants in the implementation processes of successful initiatives. Thus, both project leaders and trainers, who were part of these processes, could provide broad and deep insights on implementation determinants.

Point 2: Secondly, the study's data collection and analysis are unclear. The study only stated that semi-structured interviews were conducted via video conference using Zoom and that the interviews were recorded and transcribed verbatim. However, the study did not mention how the interviews were analyzed or provide any specific coding methods.

Response 2: Thank you for your feedback. We agree that the manuscript needed more information on interview analysis and the coding process. Therefore, we have added a brief explanation about the coding methods as follows: 

Line 238: “The textual data (the transcripts) was read and then text fragments were analysed to be associated with relevant codes. When developing the coding framework, we used a unified deductive and inductive category approach. In the case of the deductive categories, they were drawn out of the semi-structured interview guide based on the CFIR domains and constructs, while the inductive subcategories arose at the time of the interview. Thus, interviews were used to explore further subcategories to identify areas relevant to a broader scope of discussion. To ensure the coding framework was comprehensive, interviews were thoroughly reviewed multiple times. In cases in which no other categories could be extracted from the interview transcripts, a line-by-line analysis of all the interviews was carried out using the coding framework [35, 36]. 

Point 3: Finally, the study's results are not presented in a clear and organized manner. The study presents the results by CFIR domains, but the presentation is not comprehensive and may confuse the reader. It is unclear how the negative, neutral, and positive perceptions were measured or how they contribute to the results.

Response 3: Many thanks for the valuable remark. We agreed with the reviewer that it needed to be clarified how negative, neutral, and positive perceptions were measured and how this contributed to the results. Therefore, the following changes were made to the manuscript for clarification: 

I. Information was added to the methodology: 

Line 248: “Additionally, through the interviews, we quantified the total number of barriers and facilitators encountered when implementing the project according to the perceptions of the 21 participants”.

II. Table 2 title and headings were modified:

The title "Participants' perceptions regarding CFIR framework" was changed to "Barriers and facilitators for PA initiative implementation based on the CFIR". 

The heading "participants perceptions" was changed to "Number of barriers and facilitators identified according to participants perceptions". 

We believe that the changes in the title and headings of Table 2 make it clearer for the reader to understand our results since we are giving an account of how many and which were the barriers and facilitators identified according to participants' perceptions. 

III. Information was added in the results section, we included the number of barriers or facilitators encountered. For example:

Line 269: “All participants (n= 21) believed there was an advantage of implementing this project within their sports associations and almost the half (n=9) perceived that this sports project was adaptable to fill existing gaps at the local level. In addition, most of them (n=18) perceived positively the intervention source and the external institution that developed it (e.g. DOSB, DAlzG)”. 

Response to Reviewer # 4 

Point 1: First of all, the authors said "ethical approval from a committee was not sought" and this research included a patients, even there was no interventions, this kind of research also require ethical approval.

Response 1: Thank you for indicating that our study may be understood as if we included patients (persons with dementia). Although if you read the Sample and Recruitment section (line 198) closely, we specified that, the participants in this study are Project Leaders and Sports Trainers. We did not include people with dementia. The study aimed to analyse the processes behind the implementation and identify possible factors for successful project applications. Therefore, we included individuals involved in design and implementation processes rather than the target population (people with dementia).

Point 2: How the authors did not include participants that were patients? Dose the PwD not patients???

Response 2: Thank you again for your comments. Again, we reiterate that people with dementia were not included in the interviews, as clarified in the previous answer.

Point 3: "After study approval, members of the steering committee informed project staff about our study, and those interested in participating could do it voluntarily." approval from who? the authors said firstly that they did not sought ethical approval, then they said after study approval. ?????

Response 3: Again, thank you for your review and comment. Now if you read line 155 of the manuscript carefully, we specify as following: 

“… the study was presented in detail to the steering committee of the DOSB project "sports moves people with dementia" (Sport bewegt Menschen mit Demenz). This way, steering committee members decided whether to allow this study. The DOSB is a recognised entity with extensive experience in complying with the guidelines and ethical standards of its sports projects, ensuring the consent, confidentiality and anonymity of those participating”. 

Therefore, DOSB (Deutsche Olympische Sportbund), in English German Olympic Sports Confederation, was the entity responsible for approving the study, as specified in the manuscript.

---

## [Decision Letter · Decision Letter 1]

26 Jul 2023

Implementing a physical activity project for people with dementia in Germany – identification of barriers and facilitator using consolidated framework for implementation research (CFIR): A qualitative study

PONE-D-22-26436R1

Dear Dr. Cardona,

We’re pleased to inform you that your manuscript has been judged scientifically suitable for publication and will be formally accepted for publication once it meets all outstanding technical requirements.

Kind regards,

Eshak I Bahbah

Academic Editor

PLOS ONE

Reviewers' comments:

Reviewer's Responses to Questions

**Comments to the Author**

1. If the authors have adequately addressed your comments raised in a previous round of review and you feel that this manuscript is now acceptable for publication, you may indicate that here to bypass the “Comments to the Author” section, enter your conflict of interest statement in the “Confidential to Editor” section, and submit your "Accept" recommendation.

Reviewer #1: All comments have been addressed

Reviewer #4: All comments have been addressed

2. Is the manuscript technically sound, and do the data support the conclusions?

Reviewer #1: Yes

Reviewer #4: Yes

3. Has the statistical analysis been performed appropriately and rigorously? 

Reviewer #1: Yes

Reviewer #4: Yes

4. Have the authors made all data underlying the findings in their manuscript fully available?

Reviewer #1: Yes

Reviewer #4: Yes

5. Is the manuscript presented in an intelligible fashion and written in standard English?

Reviewer #1: Yes

Reviewer #4: Yes

6. Review Comments to the Author

Reviewer #1: (No Response)

Reviewer #4: (No Response)

7. PLOS authors have the option to publish the peer review history of their article (what does this mean?). If published, this will include your full peer review and any attached files.

Reviewer #1: No

Reviewer #4: No

---

## [Editor Report · Acceptance letter]

1 Aug 2023

PONE-D-22-26436R1 

Implementing a physical activity project for people with dementia in Germany – identification of barriers and facilitator using consolidated framework for implementation research (CFIR): A qualitative study 

Dear Dr. Cardona:

I'm pleased to inform you that your manuscript has been deemed suitable for publication in PLOS ONE. Congratulations! Your manuscript is now with our production department. 

Kind regards, 

on behalf of

Dr. Eshak I Bahbah 

Academic Editor

PLOS ONE